# Heart Rate and Oxygen Uptake During Recovery from High-Intensity Interval Training: A Retrospective Analysis

**DOI:** 10.3390/ijerph22070999

**Published:** 2025-06-25

**Authors:** Todd A. Astorino, Gregory C. Bogdanis, Eduardo C. Costa

**Affiliations:** 1Department of Kinesiology, California State University—San Marcos, San Marcos, CA 92096, USA; 2School of Physical Education and Sport Science, National and Kapodistrian University, 17237 Athens, Greece; gbogdanis@phed.uoa.gr; 3ExCE Research Group, Department of Physical Education, Federal University of Rio Grande do Norte, Natal 59078-970, Brazil; ecc.ufrn@gmail.com

**Keywords:** interval exercise, recovery, intermittent exercise, cardiorespiratory fitness, blood lactate, cycling

## Abstract

Background: Increases in maximal oxygen uptake (V̇O_2_max) occur with high-intensity interval training (HIIT), partially due to an extended duration spent at or near maximal V̇O_2_ or heart rate (HRmax). HIIT induces a delay in HR and V̇O_2_ during exercise, leading to consistently high HR/V̇O_2_ values in recovery between intervals. Purpose: This study compared the V̇O_2_ and HR response between exercise and recovery to various cycling HIIT protocols using data from seven prior studies. Methods: Healthy, active men and women (N = 104, age and V̇O_2_max 24 ± 5 yr and 40 ± 7 mL/kg/min) underwent HIIT protocols having different durations (30–60 s), intensities (70–85 percent of maximal workload (%Wmax), and recovery periods (10–75 s). V̇O_2_, HR, and blood lactate concentration (BLa) were assessed. Results: Across studies, peak HR was equal to 90.7 ± 6.2% HRmax. Results showed no significant difference in mean HR (159 ± 14 vs. 160 ± 15 b/min, *p* = 0.48) or V̇O_2_ (1.97 ± 0.47 vs. 1.98 ± 0.48 L/min, *p* = 0.82) between exercise and recovery. Conclusions: These data show elevated V̇O_2_ and HR during recovery from HIIT, suggesting a substantial, sustained load on the cardiovascular system in recovery from interval exercise.

## 1. Introduction

The American College of Sports Medicine recommends that all adults achieve 150 min/wk of moderate physical activity or 75 min/wk of vigorous activity to improve health and fitness status [1]. One widely reported adaptation demonstrated in response to regular physical activity is an increase in maximal oxygen uptake (V̇O_2_max) which is strongly associated with reductions in cardiovascular morbidity and mortality in adults [2,3]. Data from the HERITAGE study showed that 20 wk of moderate-intensity continuous training (MICT) led to a significant increase in V̇O_2_max in participants diverse in age, sex, ethnicity, and fitness level [4]. This response was accompanied by a significant reduction in resting and exercise heart rate (HR) and significant increase in stroke volume (SV) assessed during submaximal exercise [5], suggesting the incidence of adaptations within the cardiovascular system which enhance oxygen delivery to the working muscle.

Subsequent data in inactive adults demonstrated that higher-intensity MICT at 65–80% V̇O_2_peak elicits significantly greater increases in V̇O_2_max versus training at lower intensities (40–55% V̇O_2_peak) [6]. In addition, exercise training requiring greater time spent at or near V̇O_2_max or HRmax elicits greater increases in V̇O_2_max [7], suggesting the importance of vigorous exercise in optimizing the V̇O_2_max response to exercise training.

In the last three decades, there has been tremendous interest in the efficacy of high-intensity interval training (HIIT) for improving various health- and fitness-related outcomes similarly to MICT [8,9]. HIIT is defined as “intermittent exercise bouts performed above moderate intensity” [10] which elicit 77–95% HRmax. Although infinite permutations of HIIT exist, one widely used protocol consists of ten 1 min intervals, first used by Little et al. [11]. This regimen has been implemented in healthy adults [12] as well as those with chronic disease [13]. Prior data show that the 10 X 1 min protocol elicits peak HR and V̇O_2_ equal to 88 ± 6% HRmax and 71 ± 6% V̇O_2_max when intensity is equal to 75% Wmax [14]. Similar intensities (~86% HRmax) occur when this protocol is performed at 90 percent maximal treadmill velocity [15]. Yet, higher intensities are attained when exercise is performed at a higher workload equal to 85% Wmax (94% HRmax and 83% V̇O_2_max) [16].

It is evident that acute interval exercise places substantial stress on the cardiovascular system, represented by near-maximal values of HR and V̇O_2_. However, the structure of HIIT, consisting of relatively brief bursts (≤1 min) of exercise, leads to a lag in HR and V̇O_2_ whose magnitude of increase during each interval is not representative of the absolute exercise intensity. Consequently, the recovery period between short intervals demonstrates the maintenance of or increases in HR and V̇O_2_ [17], which is in accordance with the maintenance of stroke volume (SV) [16]. Although the acute HR and V̇O_2_ response to HIIT is well-described, few studies have explored how these variables change during the recovery period or compared recovery values to exercise interval responses. In 10 men, Farias-Junior et al. [18] described the HR, V̇O_2_, carbon dioxide production (V̇CO_2_), and minute ventilation (V̇_E_) responses to exercise and recovery periods during low-volume HIIT performed on a treadmill. Their results showed the maintenance of high V̇O_2_ (56–80% of the V̇O_2_ attained during the intervals) during the recovery period. In 11 trained cyclists, Pugh et al. [19] showed elevated V̇O_2_ in the 4.5 min recovery period following 30 s cycling sprints. It is apparent that the duration, intensity, and mode of recovery affect the energy expenditure and %HR/V̇O_2_max maintained between exercise intervals and, in turn, participant tolerance to subsequent intervals which, in turn, impacts the quality of the training session.

Due to the paucity of existing data examining responses in the recovery period from HIIT, this retrospective study aimed to compare the V̇O_2_ and HR response between exercise and recovery in response to various HIIT protocols previously performed in our laboratory by habitually active adults. A secondary aim was to examine correlates of the %V̇O_2_max response to HIIT to better understand potential variables which are related to training at high relative intensities. These data are useful to clinicians and scientists who implement HIIT protocols with a goal of optimizing the adaptive response to training. For example, if the maintenance of high HR/V̇O_2_ occurs in the recovery period from HIIT, it is possible that this factor needs to be considered when designing various HIIT regimens. Furthermore, the various benefits associated with HIIT should be explained not only by the near-maximal responses induced by these efforts but also by the high values sustained in the recovery period.

## 2. Materials and Methods

*Experimental design and participants*: All participants were informed of the procedures in accordance with the protocols and provided written informed consent, with each study approved by the California State University—San Marcos Institutional Review Board. The study was implemented in accordance with the Declaration of Helsinki. The protocol number and date of approval of the three most recent studies are denoted below, yet this information does not exist for the earlier HIIT studies: Astorino et al. (2019) [20] #1282557-1, 26 June 2018; Bogdanis et al. (2021) [21] #1326037-1, 23 October 2018; and Astorino and Emma (2021) [14] #1565370-1, 25 February 2020.

The results presented in this paper were acquired from seven studies using a within-subjects, randomized, crossover design. A total of 104 men and women (age ranging from 18 to 48 yr) underwent various low-volume HIIT protocols. All were physically active for a minimum of 12 mo and participated in non-competitive sport, resistance training, aerobic exercise, or surfing for ≥150 min/wk (duration = 6 ± 2 h/wk). They were non-obese, did not smoke, and were free from any joint issue which would preclude vigorous exercise.

Participants refrained from intense exercise for 36 h before all sessions and did not eat for a minimum of 3 h pre-trial, which were confirmed with a brief survey completed before exercise. Maximal oxygen uptake was initially determined on the cycle ergometer, and resultant values of maximal workload (Wmax) and HRmax were used to prescribe intensities for the HIIT sessions. During exercise, V̇O_2_, HR, and blood lactate concentration (BLa) were continuously acquired.

*Assessment of V̇O_2_max*: During this session, body mass and height were assessed using a scale and stadiometer to calculate body mass index (BMI). Subsequently, participants were prepared for incremental exercise on an electrically braked cycle ergometer (Velotron RacerMate, Quark, Spearfish, SD, USA), during which power output was increased in a ramp-like manner by 20–35 Watt/min after a 2 min warm-up at 40–70 Watt. Different warm-up and work-rate increments were used to accommodate the disparate sexes, body sizes, and fitness levels of our participants to elicit volitional fatigue in approximately 8–10 min. Volitional exhaustion occurred when pedal cadence was below 50 rev/min. Heart rate was determined using telemetry (Polar, Woodbury, NY, USA), and 15 s pulmonary gas exchange data (V̇O_2_, V̇CO_2_, V̇_E_, and RER) were obtained during exercise using a metabolic cart (ParvoMedics True One, Sandy, UT, USA) which was calibrated before testing following manufacturer guidelines. V̇O_2_max was identified as the mean of the two highest values in the last 60 s of exercise, and V̇O_2_max attainment was confirmed using the following criteria: change in V̇O_2_ < 0.15 L/min at V̇O_2_max; HRmax < 10 beats/minute of 220–age, and respiratory exchange ratio (RER) > 1.10 [22]. HRmax and Wmax were identified as the peak values consistent with exhaustion. Blood lactate concentration (BLa) was assessed 3 min post-exercise using a portable monitor and lancet.

*Interval exercise protocols*: Table 1 exhibits the characteristics of the HIIT protocols completed. During all protocols, gas exchange data and HR were acquired every 15 s using identical procedures to those described above for the testing of V̇O_2_max.

Across studies, the interval duration ranged from 0.5 to 1.0 min and there was a 0.16–1.25 min period of active recovery between each exercise interval. Intensity ranged from 70 to 85% Wmax to accommodate the average fitness of our participants. Protocols were low-volume, as the work duration ranged from 6 to 10 min across studies and the number of exercise intervals ranged from 8 to 12. All sessions were performed on an ergometer identical to that used for V̇O_2_max testing and required an initial warm-up for 4–5 min at low intensity ranging from 10 to 25% Wmax; then the desired %Wmax was selected and the HIIT protocol began. Target workload was fixed during exercise, and participants self-selected a suitable cadence on the cycle ergometer between 50 and 80 rev/min. Recovery intensity was maintained at the same intensity as the warm-up and ranged from 10 to 25% Wmax across studies. Verbal encouragement was provided during exercise.

*Assessment of blood lactate concentration*: Prior to exercise, after a 5 min seated rest, a 0.7 µL blood sample was taken from a fingertip using a lancet (Owen Mumford Inc., Marietta, GA, USA) and portable monitor (Lactate Plus, Sports Research Group, New Rochelle, NY, USA) to assess BLa. The fingertip was cleaned with a damp towel and dried, and then the first drop of blood was wiped away. This measure was repeated midway during exercise (for example, immediately at the end of bout 5 of the 10 X 1 protocol) and three minutes post-exercise following identical procedures.

*Analysis of oxygen uptake, HR, and blood lactate concentration data*: The mean V̇O_2_ and HR were determined as the average value from the entire bout, not including the warm-up, and were calculated from 15 s time-averaged values. V̇O_2_ and HR from each exercise interval were determined as the mean of the last two 15 s values and first value in the recovery period, with the exception of one study which only allotted a 10 s recovery period between exercise intervals. We used the first value in the recovery period as this approach was used in our prior studies to classify the HR/V̇O_2_ response to HIIT. The peak value was represented as the highest value from any exercise interval. Recovery V̇O_2_ and HR were calculated as the mean of the remaining 15 s values prior to the subsequent exercise interval, which included one (10 s recovery), three (1 min recovery), or four (1.25 min recovery) 15 s data points depending upon the duration of the recovery period. These outcomes are expressed in absolute (L/min and b/min) and relative units (%V̇O_2_max and HRmax). The BLa value reported is the highest value obtained at any point in response to the HIIT session. Energy expenditure (in kcal) was estimated using gas exchange data acquired from the metabolic cart.

*Statistical analysis*: Data are reported as means and standard deviations (SDs) and were analyzed using SPSS Version 27 (IBM, Armonk, NY, USA). The normality of data distributions was identified using the Shapiro–Wilks test. A two-way ANOVA with repeated measures was used to identify differences in HR and V̇O_2_, with one within-subjects factor (exercise interval vs. recovery = 2 levels) and one between-subjects factor (study = 7 levels). One-way ANOVA was used to identify differences in BLa across the seven studies. If a significant F ratio occurred, Tukey’s post hoc test was used to identify differences between means. The Greenhouse–Geisser adjustment was used if the sphericity assumption was violated. The dependent *t*-test was used to examine differences in mean/peak HR and V̇O_2_ between exercise and recovery. Pearson product moment correlation coefficient was used to identify pairwise relationships between variables. Cohen’s d was used as a measure of effect size, with a small, medium, and large effect equal to 0.2, 0.5, and 0.8, respectively [27]. Statistical significance was set at *p* < 0.05.

## 3. Results

*Participant physical characteristics*: The mean age, body mass, BMI, V̇O_2_max, HRmax, and Wmax in our participants (60 men and 44 women) were equal to 24 ± 5 yr, 72 ± 13 kg, 24 ± 3 kg/m^2^, 40 ± 7 mL/kg/min and 2.9 ± 0.7 L/min, 185 ± 10 b/min, and 273 ± 58 W, respectively.

*Physiological data*: Across all participants, peak intensity was equal to 78.5 ± 10.3% V̇O_2_max and 90.7 ± 6.2% HRmax, respectively, which reflects the vigorous nature of these HIIT protocols having a unique volume, intensity, number, duration, and recovery. Peak BLa and energy expenditure were equal to 9.9 ± 3.1 mM and 174 ± 46 kcal. Our data show no difference in mean HR (159 ± 14 vs. 160 ± 15 b/min, *p* = 0.48) or mean V̇O_2_ (1.97 ± 0.47 vs. 1.98 ± 0.48 L/min, *p* = 0.82) between interval exercise and the recovery period, respectively. Across studies, the mean exercise interval HR and V̇O_2_ ranged from 151 to 171 b/min and 1.62 to 2.19 L/min, which are equivalent to 82–91% HRmax and 63–72% V̇O_2_max (Table 2). Similar values were shown for HR and V̇O_2_ in the recovery period, equal to 81–92% HRmax and 62–77% V̇O_2_max. With the exception of one study [24], exercise interval and recovery HR and V̇O_2_ were similar in all studies.

For HR (b/min), repeated-measures ANOVA showed no main effect of time (exercise interval versus recovery period, *p* = 0.82, η^2^_p_ = 0.01), yet there was a significant time X study interaction (*p* < 0.001, η^2^_p_= 0.73). Post hoc analyses showed significant differences in mean HR between various studies (*p* = 0.014, Table 2). In regard to mean V̇O_2_ (L/min), no main effect of time was shown (*p* = 0.68, η^2^_p_ = 0.03), yet there was a significant time X study interaction (*p* < 0.001, η^2^_p_ = 0.60). Significantly lower mean V̇O_2_ was shown in the Astorino and Emma [14] study compared to Wood et al. (*p* = 0.006) [26], Reigler et al. (*p* = 0.002) [25], and Astorino et al. (*p* = 0.01) [20].

Blood lactate concentration significantly varied across studies (*p* < 0.001), with values ranging from 6.6 to 13.2 mM (Table 2). The lowest value occurred in response to HIIT having the lowest absolute intensity and duration (70% Wmax and 30 s) [25], whereas the highest values were derived from studies in which self-paced HIIT was performed at 80% Wmax [20,23] and another study using constant intensity equal to 85% Wmax [24].

*Correlation data*: Significant and strong relationships were shown between mean HRbout and recovery (r = 0.84, *p* < 0.001) and mean V̇O_2_bout and recovery (Figure 1). Figure 2 shows associations between various variables and %HR/V̇O_2_max from all participants. Significant correlates of peak V̇O_2_ (%V̇O_2_max) included absolute V̇O_2_max (r = −0.40, *p* = 0.004), mean HR in b/min (r = 0.33, *p* = 0.001), peak HR (%HRmax) (r = 0.61, *p* < 0.001), mean HRbout (r = 0.48, *p* < 0.001), and mean HRrecovery (r = 0.20, *p* = 0.048). For peak V̇O_2_, there was no association between mean V̇O_2_ in L/min (r = −0.04, *p* = 0.71), mean V̇O_2_bout (r = 0.12, *p* = 0.23), mean V̇O_2_recovery (r = −0.12, *p* = 0.24), or BLa (r = 0.11, *p* = 0.27). Outcomes significantly associated with peak HR (%HRmax) included absolute V̇O_2_max (r = −0.42, *p* < 0.001), meanHRbout (r = 0.65, *p* < 0.001), meanHRrecovery (r = 0.56, *p* < 0.001), and BLa (r = 0.30, *p* = 0.003).

## 4. Discussion

This retrospective analysis used results from seven studies performed in our laboratory to portray the V̇O_2_ and HR response during the recovery period from HIIT, which is poorly understood. Prior studies show significant increases in V̇O_2_max [9,10] and glycemic control [28] as well as reductions in body fat [29] and blood pressure [30] with chronic HIIT. Compared to MICT, HIIT requires a higher intensity which elicits significantly higher HR, V̇O_2_, and BLa [24], which are thought to be associated with the adaptive response. The results of the present study demonstrate elevated HR and V̇O_2_ during the recovery period from low-volume HIIT, with these values not significantly different from values induced by the exercise intervals. Overall, a substantial portion of time spent at near-maximal intensity occurs in the recovery period between intervals, and further work is needed to discern whether this contributes to the widely reported long-term adaptive response.

Our results (Table 2) show no significant difference in HR or V̇O_2_ between the exercise interval and recovery period, with some studies showing slightly higher values during the recovery period versus the exercise interval. In a prior study in young men [17], gas exchange data were acquired during a 4 X 30 s sprint interval training (SIT) session. Results showed that V̇O_2_ increased from 53 to 72% V̇O_2_max during the exercise intervals, with significantly elevated values early (10–30 s) in the active recovery period equal to 88–99% V̇O_2_max. By minute 2 of the recovery period, V̇O_2_ decreased to approximately 38% V̇O_2_max, and it further declined to 33% V̇O_2_max during minutes 3 to 4 of the recovery period. These initial recovery V̇O_2_ values are substantially higher than those reported in the present study, likely due to the “all-out” nature of SIT requiring dramatically higher power outputs which, in turn, elicit a more prolonged and severe recovery period than HIIT. In addition, V̇O_2_ was recorded breath by breath, which augments the V̇O_2_ response compared to longer time-averaging intervals [31]. In a more recent study, Ksoll et al. [32] required 24 men (V̇O_2_max = 54 mL/kg/min) to complete two work-matched HIIT protocols: five sets of 3 min intervals at 80% V̇O_2_max followed by a 3 min recovery period, or five sets of six 30 s intervals at 80% V̇O_2_max followed by a 30 s recovery period. Despite equivalent mean V̇O_2_ between protocols, 3 min intervals elicited a significantly higher V̇O_2_ and total duration above 80% V̇O_2_max than the 30 s intervals, which exhibited significantly higher V̇O_2_ in the recovery period. In addition, HR and cardiac output (CO) were significantly higher between recovery from 30 s and 3 min intervals. Similar findings were observed by Farias-Junior et al. [18] in comparing HR and gas exchange data during running-based HIIT sessions with different work–recovery durations (10 X 1/1 min vs. 20 X 30/30 s) at 100% maximal velocity interspersed with passive recovery. The 1/1 min HIIT protocol elicited a greater amplitude (i.e., work–recovery differences) in physiological responses compared to the 30/30 s HIIT protocol.

Stanley and Buchheit [33] characterized the V̇O_2_ and stroke volume (SV) response to 3 X 3 min HIIT at 90% Wmax, interspersed with 2 min of active recovery at 30 or 60% Wmax, in trained cyclists. During the recovery period, results showed the maintenance of relatively high HR (~80 and 85% HRmax at 30 and 60% Wmax) and V̇O_2_ (~60 and 80% V̇O_2_max at 30 and 60% Wmax) as well as elevations in SV and CO. This sustained elevation in V̇O_2_ and hence CO throughout the entire exercise protocol, including the exercise intervals and recovery period, likely potentiates adaptations in slow-twitch muscle fibers and myocardial growth and, in turn, contributes to increases in V̇O_2_max when completed long-term [34]. Moreover, as acute intensity of exercise has been identified as a key regulator of mitochondrial biogenesis [35], and a maintenance of high V̇O_2_ in the recovery period, despite a relatively low external power output, may partially elicit these responses. Together, this elevation in V̇O_2_ and HR in the recovery period may be important to elicit an increase in V̇O_2_max in response to exercise training.

There are several possible explanations for the relatively high values of V̇O_2_ and HR in the recovery period reported in the present study. First, there is a marked delay in the V̇O_2_ response to short exercise intervals; moreover, V̇O_2_ is significantly lower in response to short versus long intervals at 100% Wmax when intensity and total work are matched [18,36]. Second, there are slower O_2_ kinetics observed in response to cycling versus running [37]. Third, nonathletic adults, such as those used in the present study, have slower O_2_ kinetics which may attenuate the V̇O_2_ gain during each exercise interval and cause V̇O_2_ to rise early in the recovery period. Fourth, the completion of each exercise interval accelerates O_2_ kinetics [38] and, combined with the V̇O_2_ slow component [39], may lead to elevated V̇O_2_ in the recovery period.

The importance of substantial elevations in V̇O_2_ and HR in the recovery period between intervals, as shown by our results and others, is relatively unclear due to the paucity of prior results revealing the potential impact of recovery V̇O_2_ on subsequent changes in health- and fitness-based outcomes. Prior data in trained cyclists [33] showed similar SV values in the recovery period from HIIT to those during the exercise intervals, causing the authors to conclude that part of the circulatory response to HIIT may be due to hemodynamic changes occurring in the recovery period allowing the maintenance of a high CO. In active men and women, Coe and Astorino [15] revealed the maintenance of maximal SV throughout 10 X 1 min and 4 X 4 min HIIT, yet this was not a training study, so no long-term changes in V̇O_2_max were assessed. In overweight men, Boyd et al. [40] compared changes in V̇O_2_max and cycling performance in response to nine sessions of HIIT requiring 1 min efforts at 70 or 100% Wmax. Training at 100% Wmax, which would likely elicit higher V̇O_2_ in each exercise interval and in the recovery period, revealed substantially greater increases in V̇O_2_max (28 vs. 11%) and cycling performance (14 vs. 9%) versus lower-intensity HIIT. Nevertheless, another study in inactive women [12] showed no significant differences in the V̇O_2_max response to 12 wk of HIIT at 60–80% or 80–90%Wmax. Further study is needed to monitor HR and V̇O_2_ in the recovery period from HIIT and identify their potential association with training-induced changes in V̇O_2_max and exercise performance.

Active recovery is typically prescribed after HIIT to accelerate metabolic recovery (BLa and H+ removal) and maintain a certain level of V̇O_2_ to expedite the attainment of near-maximal V̇O_2_ in successive intervals [34,40,41]. Nevertheless, it is apparent that active recovery may reduce exercise tolerance in subsequent intervals [34] despite allowing a substantial duration to be spent at or near V̇O_2_/HRmax [34]. Overall, the intensity and duration of recovery depend on the interval number, intensity, and duration, and it is likely that the specific characteristics of the recovery period can be manipulated to address different participant goals such as augmenting time at or near V̇O_2_max, the contribution of nonoxidative metabolism, and energy expenditure. For example, in the Olney et al. [24] study, V̇O_2_ in the recovery period was substantially lower than the exercise interval value due to the slightly longer recovery period (75 vs. 60 s) employed, which led to a greater decline in V̇O_2_ compared to the other studies having a shorter recovery period. Yet this longer recovery period was afforded to reduce fatigue and preserve performance, which is important in participants having average V̇O_2_max, such as the majority of participants enrolled in our studies.

Figure 1 shows strong, significant associations between exercise and recovery HR and V̇O_2_, supporting our data showing no significant difference in these outcomes. In addition, our results (Figure 2) show significant inverse associations between %V̇O_2_max and absolute V̇O_2_max, suggesting that individuals having lower cardiorespiratory fitness tend to exercise at higher fractions of %V̇O_2_max during HIIT than more-fit adults. Although speculative, the greater training-induced increase in V̇O_2_max observed in less-fit adults [8,9] may be partially associated with the higher V̇O_2_ and HR attained during the exercise interval and the recovery period. Nine male participants had V̇O_2_max ≥ 50 mL/kg/min, and their mean intensity attained during the exercise intervals was equal to 66.5% V̇O_2_max (57–70% V̇O_2_max). In contrast, 12 female participants with V̇O_2_max ≤ 31 mL/kg/min attained intensity equal to 86.8% V̇O_2_max (75–97% V̇O_2_max) during HIIT. Participants with lower V̇O_2_max likely exhibit a lower ratio of slow- to fast-twitch fibers [42] and a ventilatory or lactate threshold at lower relative intensities, in turn requiring a larger contribution of nonoxidative metabolism toward energy expenditure supporting interval exercise. It is recommended that in fit adults, interval protocols requiring intensities well above 85% Wmax, or durations > 1 min, are needed to optimize time spent at higher fractions of %V̇O_2_max. Another strategy to maximize the V̇O_2_ response in this population would be to reduce recovery duration and blunt the decline in V̇O_2_ seen with a longer recovery period.

This study faces several limitations. First, data were combined across all studies despite small differences in HIIT structure, intensity, and recovery, which limits the generalizability of our results. Although %Wmax varied across studies, peak BLa (6.6–13.2 mM) and relative intensity (83–92% HRmax) are representative of near-maximal exercise characteristic of HIIT, suggesting that methodological discrepancies have minimal impact upon our conclusions. The Olney et al. study [24] revealed similar HR and V̇O_2_ responses between high- (6 X 2 min at 70% Wmax) and low-volume HIIT (8 X 1 min at 85% Wmax), suggesting a similar cardiorespiratory load despite different exercise volumes, intensities, and recovery periods. Second, the results were acquired from young, healthy, habitually active adults, so data cannot be applied to other populations. The mean V̇O_2_max ranged from 37 to 42 mL/kg/min across studies, which represents average cardiorespiratory fitness for young men and women [43]. The relative similarity in participant V̇O_2_max across studies suggests a minimal impact of fitness on our results. Third, these results only apply to cycling-based HIIT, and different results may occur when treadmill [21,44] or upper-body interval exercise [14,45] is performed. Fourth, in all studies, we implemented active recovery at low intensities, and different results could occur in response to passive recovery or different modes or intensities of active recovery. Lastly, HR and V̇O_2_ estimates calculated from each exercise interval included the first value in the recovery period, per methods used in our prior work. Nevertheless, if this data point was not used, it is likely that the exercise interval V̇O_2_ would have been substantially lower than values acquired in the recovery period. However, this does not change the fact that V̇O_2_ in the recovery period comprises a substantial portion of the total oxygen consumption and duration spent at or near V̇O_2_max inherent to HIIT. Yet, this study is strengthened by the large sample of diverse sexes and fitness levels, the use of standardized procedures for the assessment of HR/V̇O_2_max, and the identical methods used to analyze the data.

## 5. Conclusions

In conclusion, our results from a large sample of adults undergoing various low-volume HIIT regimens reveal no difference in V̇O_2_ or HR between the exercise interval and subsequent recovery period. This suggests that the active recovery period between intervals maintains a substantial stress on the cardiovascular system. The impact of this additional cardiovascular load during recovery periods on cardiovascular and/or fitness-related adaptations is unclear and deserves further investigation.

## Figures and Tables

**Figure 1 ijerph-22-00999-f001:**
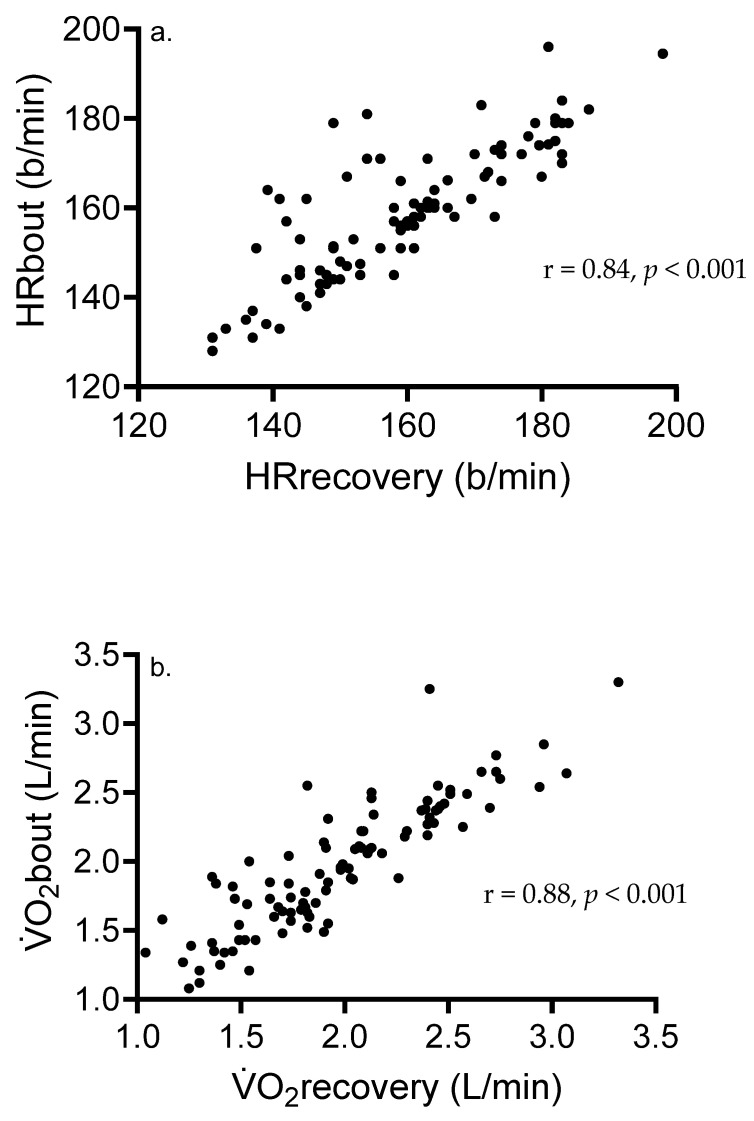
Associations between (**a**) mean HRbout and mean HR in the recovery period and (**b**) mean V̇O_2_bout and mean V̇O_2_ in the recovery period.

**Figure 2 ijerph-22-00999-f002:**
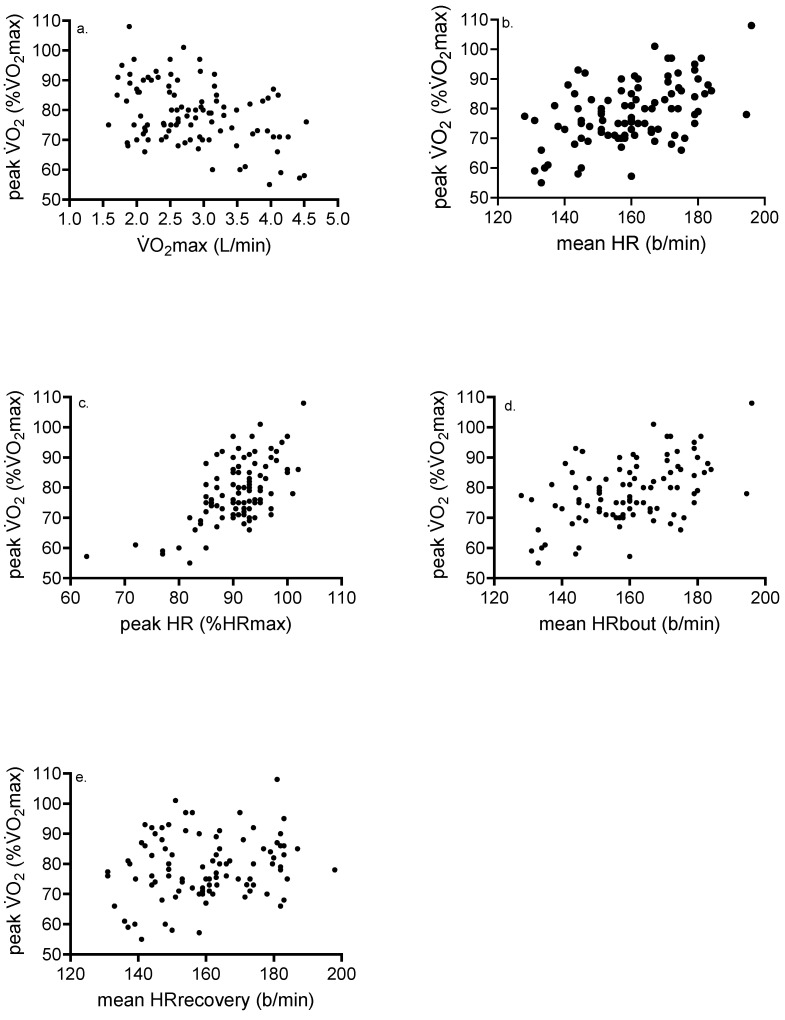
Associations between (**a**) peak V̇O_2_ (%V̇O_2_max) and V̇O_2_max, r = −0.40, *p* = 0.004; (**b**) peak V̇O_2_ (%V̇O_2_max) and mean HR, r = 0.33, *p* = 0.001; (**c**) peak V̇O_2_ (%V̇O_2_max) and peak HR (%HRmax), r = 0.61, *p* < 0.001; (**d**) peak V̇O_2_ (%V̇O_2_max) and mean HRbout, r = 0.48, *p* < 0.001; and (**e**) peak V̇O_2_ (%V̇O_2_max) and mean HRrecovery, r = 0.20, *p* = 0.048.

**Table 1 ijerph-22-00999-t001:** High-intensity interval training protocols completed in the present study.

Study	Subjects and Age (yr)	V̇O_2_max (mL/kg/min)	HRmax (b/min)	HIIT Protocol	Intensity	Recovery Period (min)	Recovery Period (%Wmax)
Astorino and Emma [14]	23 M/W 25 ± 6	37 ± 6	185 ± 12	10 X 1 min	75% Wmax	1.0	10
Astorino et al. [20]	17 M/W 26 ± 6	39 ± 4	186 ± 8	10 X 1 min	~79% Wmax	1.0	10
Bogdanis et al. [21]	5 M/W 23 ± 4	40 ± 8	183 ± 3	10 X 1 min	VT + 20%	1.0	20
Kellogg et al. [23]	14 M/W 24 ± 3	42 ± 9	188 ± 8	8 X 1 min	80% Wmax	1.0	10
Olney et al. [24]	19 M/W 24 ± 3	40 ± 6	188 ± 8	8 X 1 min	85% Wmax	1.25	20
Reigler et al. [25]	14 M/W 25 ± 8	40 ± 6	185 ± 12	12 X 30 s	70% Wmax	0.16	20
Wood et al. [26]	12 M/W 24 ± 6	41 ± 4	179 ± 10	8 X 1 min	85% Wmax	1.0	25

M = men; W = women; V̇O_2_max = maximal oxygen uptake; HR = heart rate; b/min = beats per minute; %Wmax = percent maximal workload; min = minute; s = seconds.

**Table 2 ijerph-22-00999-t002:** Mean heart rate (HR), oxygen uptake (V̇O_2_), and blood lactate response to interval exercise and recovery (mean ± SD and minimum to maximum value).

Study	HRmean_bout_ (b/min)	HRmean_recovery_ (b/min)	V̇O_2_mean_bout_ (L/min)	V̇O_2_mean_recovery_ (L/min)	BLa (mM)
Astorino and Emma [14]	151 ± 14	155 ± 14	1.62 ± 0.32	1.66 ± 0.28	7.6 ± 2.7
	128–172	131–183	1.27–2.47	1.22–2.18	3.6–12.5
Astorino et al. [20]	165 ± 14 ^a^	171 ± 14 ^a^	2.01 ± 0.41 ^a^	2.17 ± 0.47 ^a^	13.2 ± 1.7 ^a^
	141–195	147–198	1.43–2.65	1.47–3.07	9.8–15.6
Bogdanis et al. [21]	166 ± 10	168 ± 10	2.12 ± 0.49	2.19 ± 0.52	9.2 ± 2.9
	151–174	156–181	1.54–2.65	1.49–2.73	5.0–12.3
Kellogg et al. [23]	157 ± 17	163 ± 17	1.90 ± 0.50	2.05 ± 0.47	11.3 ± 3.2 ^a^
	131–179	133–184	1.08–2.55	1.25–2.70	6.8–15.2
Olney et al. [24]	171 ± 12	153 ± 13	2.10 ± 0.50	1.75 ± 0.41	9.8 ± 3.1 ^b^
	151–196	137–181	1.34–3.25	1.04–2.41	3.8–15.3
Reigler et al. [25]	159 ± 15	160 ± 15	2.19 ± 0.44 ^a^	2.23 ± 0.44 ^a^	6.6 ± 2.1 ^b^
	137–182	137–187	1.43–2.85	1.57–2.96	3.6–9.7
Wood et al. [26]	154 ± 9	153 ± 9	2.19 ± 0.40 ^a^	2.17 ± 0.42 ^a^	11.3 ± 3.3 ^a^
	144–164	144–170	1.70–3.30	1.64–3.32	6.7–17.0

HR = heart rate; V̇O_2_ = oxygen uptake; BLa = blood lactate concentration; ^a^ = *p* < 0.05 vs. Astorino et al. (2021) [14]; ^b^ = *p* < 0.05 vs. Astorino et al. (2019) [20].

## Data Availability

All pertinent data concerning this manuscript is presented here.

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
