# Peer review of "Heart Rate and Oxygen Uptake During Recovery from High-Intensity Interval Training: A Retrospective Analysis"

_ijerph, 2025, doi:10.3390/ijerph22070999_

Round 1
Reviewer 1 Report
Comments and Suggestions for Authors
This manuscript compared the V̇O2 and HR response between exercise and recovery to various cycling HIIT protocols using data from seven prior studies. The focus on recovery HR and V̇O2 responses to HIIT fills a notable gap in the literature, where the majority of research emphasizes during-exercise responses. There are limited studies that provide novel insights into a neglected aspect of HIIT physiology. Drawing attention to recovery-phase cardiorespiratory responses can open the door for future research on optimizing HIIT for fitness, health, and performance. The manuscript is scientifically sound and presents a valuable retrospective analysis of physiological responses to HIIT. However, significant concerns exist regarding the method. Below is a detailed academic review
The problem the study addresses and its contributions to the literature and practice should be clearly explained.
The number and date of the ethics committee report should be stated.
The age range of the participants is very high. This will affect the physiological responses to the exercise load. Therefore, the age distributions of the participants should be stated in the exercise protocols.
The procedures of the study are generally mentioned in the experimental design section of the study. In this section, the measurements taken before starting the study, the study performed later, and the measurements taken during the high-intensity loading and recovery sessions in between should be mentioned in detail.
It should be fully explained how the seven different HIIT workouts used in this study were selected, and which characteristics of the participants were distributed to these protocols. The descriptive statistics of the participants in each protocol should also be stated.
In Line 126, please state the lactate values ​​taken during the exercise (after each interval loading?) in more detail.
In the discussion section, first explain the purpose of this study, not the background of the study, and then your own results.
There are a huge number of cited references that belong to the authors (Refs.
9,12,14,15,16,20,25,30,44,35,43,29)in the manuscript. Although some of the authors' studies are related to the study conducted, there are many studies by different authors that contribute to this field. This situation is not ethical and against the journal's writing rules. It is recommended that authors only cite minimal studies that are directly related to the study conducted.
Minor
Title: "Heart rate and oxygen uptake in recovery to high intensity interval training"
Suggestion: Use “during recovery from” instead of “in recovery to”.
Author Response
Author Rebuttal to IJERPH-3679005 manuscript; Heart rate and oxygen uptake during recovery to high intensity interval training
General comments: We thank the Reviewers for thorough and helpful comments concerning this paper, which discusses a mostly ignored topic in the interval training literature, which is examining the changes in cardiometabolic responses in recovery during various high intensity interval training (HIIT) protocols. Our results from 7 studies employing different low-volume HIIT protocols show maintenance of high heart rate (HR) and oxygen uptake (VO2) during recovery which is not significantly different from the in-task values. Together, these results suggest that the near-maximal HR/VO2 values associated with acute HIIT also include the recovery periods in which exercise intensity is low. These results will be of interest to scientists and clinicians who employ HIIT and strive to implement the most optimal protocols to elicit various health and fitness related benefits in their clients. Modifications have been made to the submission in red font, and please note that some line numbers have changed in this new iteration. Overall, we hope that the revised paper and our point-by-point rebuttal below have satisfied all concerns held by the Reviewers, and we are confident that this new iteration of our work is of sufficient quality to merit publication in your Journal.
____________________________________________________________________________
Specific comments: The responses to each reviewer are appended below and we hope that these responses satisfy the Reviewers’ concerns.
Reviewer 1; We thank this reviewer for highlighting the novelty of our work, its value, as well as for praising the soundness of our submission and its design. Below, we have responded to each concern raised by this reviewer.
Study problem and contribution to the literature: This is a great point you make, and new text has been added to ln 81 85 to further explain this aspect, and we hope that this addition satisfies your concern.
Number and date of the ethical approval of these studies: All studies were submitted for Human Subjects Review and approved by the CSU—San Marcos Institutional Review Board, which is stated in ln 85 of this paper. Unfortunately, many of these studies were completed so long ago, that no record of this information remains, since our campus only transferred to on-line management of research protocols in late 2018. In the text, we have added this information for the more recent studies (3) which contain this information. We hope that this added text and explanation appease your fair concern.
Participant age range: This is a fair comment you make, yet the majority of participants in these studies were between 18 – 30 yr old, with only a few instances of participants in their 40s. In fact, the oldest participant in each study was the First Author, whose VO2max is very similar to mean values shown by our participants of approximately 40 mL/kg/min. We acknowledge that age does impact the HR response to exercise as it is typically attenuated in older compared to younger adults, yet it is our belief that aerobic fitness has a more dramatic impact on these responses reported in this submission. Please note that prior work (Storen et al. 2017) shows no effect of age on the VO2max response to 8 wk of low-volume HIIT, suggesting minimal effect of age on responses to HIIT. Per your comment, the mean/SD age of all participants has been added to our Table, and we hope that this addition appeases your fair concern.
Detail in Methods: We appreciate your comment here, and acknowledge that this text needs to have sufficient detail to allow the reader to better understand our protocols. That said, we do not believe that a comprehensive description of all protocols used is needed here, especially considering that this content has already been published in each original article, and moreover, all elements of the HIIT protocols including intensity, duration, recovery, and number of intervals are denoted in Table 1. In addition, there is relevant text describing these protocols in ln 123 – 131. Per your comment, additional text was added to the Method to be more transparent with some elements of the Methods, and we hope that this addition appeases your fair concern.
Seven different HIIT regimens and participant characteristics: We appreciate this comment, and text has been added here to be more clear. Yet, it is not clear to the Authors what is meant by “how the seven different HIIT workouts used in this study were selected.” As our text states in ln 124 – 126, all are low-volume HIIT protocols having a total exercise volume of < 10 min, with intensity ranging from 70 – 85 %Wmax. Low-volume HIIT protocols stem from the seminal work of Dr. Jon Little, who employed 10 X 1 min intervals at a workload equal to 100 %Wmax in active adults to portray muscle metabolic responses to exercise. Our regimens align with his work in terms of volume and recovery duration, yet our intensities are lower to accommodate the lower VO2max of our participants. Please note that Table 1 includes the sample size of each study, the gender composition, as well as the mean age of participants and their VO2max and HRmax. In our view, these results represent the demographic traits of our participants and along with text presented in ln 95 - 100, we believe that this content adequately describes these individuals. We hope that this explanation and revision appease your concern.
Ln 126: Text has been added to further describe the measurement of BLa during exercise. Yet, the only BLa value reported is the ‘peak’ value which was the highest value recorded during each exercise interval, which typically was acquired post-exercise. We hope this clarification appeases your concern.
Discussion, first line: This is a great point you make, and new text has been added to ln 238 – 240 to restate the study aim and overall, rewrite this portion of the Discussion; thank you.
Author self-citation: We respect your point here, and per your comment, have reduced the number of articles coming from the 3 authors cited which are not directly related to the data presented in the paper. Consequently, we have cited other scientists’ work in this area. Thank you and we hope that this revision satisfies your concern.
Title: Per your comment, the title was revised and thank you for this suggestion.
Reviewer 2 Report
Comments and Suggestions for Authors
Heart rate and oxygen uptake in recovery to high intensity interval training
Major concerns:
- Title. Since this an analysis of previous published studies. The title should be something like: “Heart rate and oxygen uptake in recovery period during high intensity interval training in active men and women: a retrospective analysis”.
- Perhaps ideal to use the term “recovery period’ rather than just the term “recovery” and use the term “exercise interval” when implying to the period of exercising during the exercise period.
- Table 2. Data as percentage of HRmax and percentage of VO2max should also be included here.
- Fgure1 and Figure 2. The correlations and p values should be written within the figures for the ease of the reader.
- Some of the figures are not so useful in explaining the aim of your study such as Figure 3 a,b,c, d and Figure 4a,b. Please delete them if you do not use them in your Discussion section.
Minor issues:
- How was Wmax determined during the VO2max test. Was it at time of termination of test or when exhaustion?
- Line 248. At what intensity was the 5 set of six 30 s intervals performed.
- Line 311. “those enrolled in our study”. Please be specific as to which study you are referring to.
- Line 315. Use “training-induced VO2max” – to indicate the improvements in VO2max after a training period.
- Line 252. “between” rather than “from”.
Author Response
Author Rebuttal to IJERPH-3679005 manuscript; Heart rate and oxygen uptake during recovery to high intensity interval training
General comments: We thank the Reviewers for thorough and helpful comments concerning this paper, which discusses a mostly ignored topic in the interval training literature, which is examining the changes in cardiometabolic responses in recovery during various high intensity interval training (HIIT) protocols. Our results from 7 studies employing different low-volume HIIT protocols show maintenance of high heart rate (HR) and oxygen uptake (VO2) during recovery which is not significantly different from the in-task values. Together, these results suggest that the near-maximal HR/VO2 values associated with acute HIIT also include the recovery periods in which exercise intensity is low. These results will be of interest to scientists and clinicians who employ HIIT and strive to implement the most optimal protocols to elicit various health and fitness related benefits in their clients. Modifications have been made to the submission in red font, and please note that some line numbers have changed in this new iteration. Overall, we hope that the revised paper and our point-by-point rebuttal below have satisfied all concerns held by the Reviewers, and we are confident that this new iteration of our work is of sufficient quality to merit publication in your Journal.
Reviewer #2: We thank this reviewer for helpful comments concerning our submission, and below, we have responded to all points raised in his/her review.
Title: This is a great comment and per your point, the title has been slightly modified; thank you.
Terminology: This is a fair point you denote, so the terms recovery period and exercise interval have been used where applicable in our manuscript, and we hope that this revision makes these terms more transparent to the reader. Thank you for this helpful suggestion to improve the clarity of our work.
Table 2: Per your comment, these data were added in text on pg 5, ln 193 and we hope that this addition is satisfactory. Unfortunately, adding these results to the table is not possible, as the table would not adequately fit on the page with 4 new columns of results.
Figures 1 and 2: Per your comment, requisite r and p values have been included here for Figure 1; for Figure 2, they were placed in the Figure legend to reduce clutter within each figure—we hope this addition satisfies your fair concern.
Figure 3: We appreciate your valid point and agree that this particular Figure does not add much to the paper, so per your comment, we have elected to remove this from the manuscript. Thank you and we hope that this change is satisfactory.
Minor issues;
Identification of Wmax: This is a great comment you raise, as different methods used to identify Wmax can lead to disparate intensities of training performed by participants. In all of our studies, Wmax was identified as the exact PO attained at exercise termination, when pedal cadence is < 50 rev/min. This is shown in ln 122 123 of our Method, and we hope that this clarification appeases your concern.
Ln 248 intensity of the Ksoll et al. study: Thank you for this comment here; all bouts in this study were performed at 80 %VO2max, and this fact has been clarified in our text.
Ln 252: Per your comment, the word choice has been revised here, and we thank you for this suggestion.
Ln 311 ‘those enrolled in our study: Thank you for your comment here, and this text has been rewritten to be more clear.
Ln 315 ‘training-induced’: We thank you for your helpful comment and it has been inserted into our manuscript.
Round 2
Reviewer 1 Report
Comments and Suggestions for Authors
I congratulate the authors for the additions and corrections they have made, and the work has now acquired a more academic structure.